# Large-Scale Unsupervised Object Discovery

**Huy V. Vo[1,2]**  **Elena Sizikova[3]**  **Cordelia Schmid[1]**  **Patrick Pérez[2]**  **Jean Ponce[1,3]**

[1]INRIA, Département d'informatique de l'ENS, ENS, CNRS, PSL University, Paris, France
[2]Valeo.ai  [3]Center for Data Science, New York University
{van-huy.vo, cordelia.schmid, jean.ponce}@inria.fr
es5223@nyu.edu  patrick.perez@valeo.com

## Abstract

Existing approaches to unsupervised object discovery (UOD) do not scale up to large datasets without approximations that compromise their performance. We propose a novel formulation of UOD as a ranking problem, amenable to the arsenal of distributed methods available for eigenvalue problems and link analysis. Through the use of self-supervised features, we also demonstrate the first effective fully unsupervised pipeline for UOD. Extensive experiments on COCO [42] and OpenImages [35] show that, in the single-object discovery setting where a single prominent object is sought in each image, the proposed LOD (Large-scale Object Discovery) approach is on par with, or better than the state of the art for medium-scale datasets (up to 120K images), and over 37% better than the only other algorithms capable of scaling up to 1.7M images. In the multi-object discovery setting where multiple objects are sought in each image, the proposed LOD is over 14% better in average precision (AP) than all other methods for datasets ranging from 20K to 1.7M images. Using self-supervised features, we also show that the proposed method obtains state-of-the-art UOD performance on OpenImages[1].

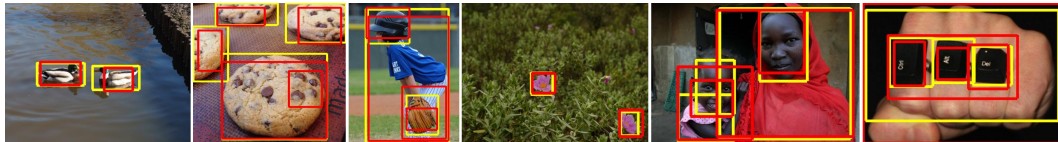

Figure 1: Sample UOD results obtained by LOD on the OpenImages dataset [35] which contains 1.7M images. Ground-truth boxes are shown in yellow, and predictions are in red. Best viewed in color.

## 1  Introduction

This paper addresses the problem of identifying prominent objects in large image collections without manual annotations, a process known as unsupervised object discovery (UOD). Early approaches to UOD focused mostly on finding clusters of images featuring objects of the same category [22, 58, 61, 62, 64, 69]. Some of them [58, 61, 62] also output object locations in images, but their evaluations are limited to small datasets with distinctive object classes. More recent techniques [8, 66, 67] focus on the discovery of image links and individual object locations within much more diverse image collections. They typically rely on combinatorial optimization to select objects, or rather, object bounding boxes, among thousands of candidate region proposals [46, 65, 67, 83] given similarity scores computed for pairs of proposals associated with different images. Although these techniques achieve promising results, their computational cost and inherently sequential nature limit the size of the dataset they can be applied to. Attempts to scale up the state-of-the-art approach [67] by reducing the search space size have revealed that this compromises its ability to discover multiple objects in

---

[1]Our code is publicly available at https://github.com/huyvvo/LOD.

each image. Other approaches to UOD focus on learning image representations by decomposing images into objects [5, 12, 23, 43, 47]. These techniques do not scale up (yet) to large natural image collections, and focus mostly on small datasets containing simple shapes in constrained environments.

Table 1: Large-scale object discovery performance and comparison to the state of the art on COCO [42] (C120K), OpenImages [35] (Op1.7M) and their respective subsets C20K and Op50K, in three standard metrics. Using VGG16 features [60], the proposed method LOD achieves top performance in both single and multi-object discovery, and scales better to 1.7M images in Op1.7M than the previous state of the art [67]. When running with self-supervised features (LOD + Self [18]), it yields the best results on Op1.7M, showing the first effective fully unsupervised pipeline for UOD. See Sec. 4 for more details.

| Method | Single-object | | | | Multi-object | | | | | | | |
| | CorLoc | | | | AP50 | | | | AP@[50:95] | | | |
| | C20K | C120K | Op50K | Op1.7M | C20K | C120K | Op50K | Op1.7M | C20K | C120K | Op50K | Op1.7M |
|---|---|---|---|---|---|---|---|---|---|---|---|---|
| EB [83] | 28.8 | 29.1 | 32.7 | 32.8 | 4.86 | 4.91 | 5.46 | 5.49 | 1.41 | 1.43 | 1.53 | 1.53 |
| Wei [71] | 38.2 | 38.3 | 34.8 | 34.8 | 2.41 | 2.44 | 1.86 | 1.86 | 0.73 | 0.74 | 0.6 | 0.6 |
| Kim [32] | 35.1 | 34.8 | 37.0 | - | 3.93 | 3.93 | 4.13 | - | 0.96 | 0.96 | 0.98 | - |
| Vo [67] | **48.5** | 48.5 | 48.0 | 47.8 | 5.18 | 5.03 | 4.98 | 4.88 | 1.62 | 1.6 | 1.58 | 1.57 |
| Ours (LOD+Self [18]) | 41.1 | 42.4 | **49.5** | **49.4** | 4.56 | 4.90 | 6.37 | **6.28** | 1.29 | 1.37 | 1.87 | **1.86** |
| Ours (LOD) | **48.5** | **48.6** | 48.1 | 47.7 | **6.63** | **6.64** | **6.46** | 6.28 | **1.98** | **2.0** | **1.88** | 1.83 |

It is natural to cast unsupervised object discovery (UOD) as the task of finding repetitive visual patterns in image collections. Recent approaches (e.g., [66, 67]) to UOD formulate it as a combinatorial optimization problem in a graph of images, selecting simultaneously image pairs that contain similar objects and region proposals that correspond to objects, with the corresponding computational limitations. The motivation behind our work is to formulate UOD as a simpler graph-theoretical problem with a more efficient solution, where objects correspond to well-connected nodes in a graph whose nodes are region proposals (instead of images in [66, 67]), and edges are weighted by region similarity and objectness. In this scenario, finding object-proposal nodes is now a ranking problem where the goal is to rank the nodes based on how well they are connected in the graph. From another perspective, ranking is rather a natural modelization choice for UOD as in our context, discovering objects means finding the most object-like regions in a set of initial region proposals which naturally amounts to ranking them according to their "objectness". As a result, a large array of methods available for eigenvalue problems and link analysis [48] can be applied to solve UOD on much larger datasets than previously possible (Fig. 1). We consider three variants of this approach: the first one re-defines the UOD objective [66, 67] as an eigenvalue problem on the graph of region proposals, the second variant explores the applicability of PageRank [4, 48] for UOD, and the final variant combines the other two into a hybrid algorithm, dubbed LOD (for large-scale object discovery), which uses the solution of the eigenvalue problem to personalize PageRank. LOD offers a fast, distributed solution to object discovery on very large datasets. We show in Sec. 4.1 and Table 1 that its performance is comparable or better than the state of the art in the single object discovery setting for datasets of up to 120K images, and over 37% better than the only algorithms we are aware of that can handle up to 1.7M images. In the multi-object discovery setting, LOD significantly outperforms all existing techniques on datasets from 20K to 1.7M images. While LOD does not explicitly address discovering relationships between images (e.g., grouping images into classes), we demonstrate that categories can be discovered as a post-processing step (see Sec. 4.2). The best performing approaches to UOD so far all use *supervised* region proposals and/or features. We also demonstrate for the first time in Sec. 4.1 that self-supervised features can give good UOD performance. Our main contributions can be summarized as follows:

- We propose a new formulation of UOD as a ranking problem, allowing the application of parallel and distributed link analysis methods [4, 48].

- We scale UOD up to datasets 87 times larger than those considered in the previous state of the art [67]. Our novel LOD algorithm outperforms others on medium-size datasets by up to 32%.

- We propose to use self-supervised features for UOD and show that LOD, combined with these features, offers a viable UOD pipeline without any supervision whatsoever.

- We conduct extensive experiments on the COCO [42] and OpenImages [35] datasets to empirically validate our method. We also demonstrate applications of our approach to object category discovery and retrieval, outperforming other existing unsupervised baselines on both tasks by a large margin.

## 2 Problem statement and related work

### 2.1 Problem statement

Consider a collection of $n$ images, each equipped with a set region proposals [66, 67]. For the sake of simplicity, we assume in this presentation that all images have exactly $r$ region proposals. We wish to find which ones of these correspond to objects, and link images that contain similar objects, without any information other than how similar pairs of proposals are. This problem is known as unsupervised object discovery (UOD) and can be formulated as an optimization problem [67] over a graph where images are represented as nodes. Let $e_{pq} \in \{0, 1\}$ for $p, q = 1, 2, \ldots, n$ be a set of binary variables indicating if two images are connected in the graph, with $e_{pq} = 1$ when images $p$ and $q$ share similar visual content. Similarly, let $x_p^k \in \{0, 1\}$ for $p = 1, 2, \ldots, n$ and $k = 1, 2, \ldots, r$ be indicator variables such that $x_p^k = 1$ when region proposal $k$ of image $p$ is an object-like region in image $p$ that is similar to an object-like region in one of the neighbors of image $p$. Let also $x_p$ be $(x_p^1, \ldots, x_p^r)^T$, $x$ be the $n \times r$ matrix whose rows are $x_p$ for $p = 1, 2, \ldots, n$ and $e$ be the binary adjacency matrix of the image graph. Then, the object discovery problem can be formulated as a combinatorial maximization problem:

$$\max_{x, e} \sum_{p=1}^{n} \sum_{q \in \mathcal{N}(p)} e_{pq} x_p^T S_{pq} x_q \quad \text{s.t.} \sum_{k=1}^{r} x_p^k \leq \nu \text{ and } \sum_{q \neq p} e_{pq} \leq \tau \quad \forall 1 \leq p \leq n, \quad \text{(C)}$$

where $S_{pq} \in \mathbb{R}^{r \times r}$ is a matrix whose entry $S_{pq}^{k\ell} \geq 0$ measures the similarity between region $k$ of image $p$ and region $\ell$ of image $q$ as well as the saliency of the respective regions, $\mathcal{N}(p)$ is a set of potential high-similarity neighbors of image $p$, and $\nu$ and $\tau$ are predefined constants corresponding to the maximum number of objects in an image and the maximum number of its neighbors, respectively. Previous approaches [66, 67] to UOD solve a convex relaxation of (C) in the dual domain and/or use block-coordinate ascent on its variables $x$ and $e$. The similarity scores $S_{pq}^{k\ell}$ are typically computed using the Probabilistic Hough Matching (PHM) algorithm from [8], which combines local appearance and global geometric consistency constraints to compare pairs of regions. A high PHM score between a pair of proposals is an indicator of whether the corresponding two proposals may correspond to a common foreground object. We follow this tradition and also use PHM scores (Sec. 4).

The objective of UOD as formulated in (C) is to find both the objects (variables $x_p^k$) and the edges linking the images that contain them (variables $e_{pq}$). Its combinatorial nature makes it hard to scale up to large values of $n$ and $r$. [67] uses a block-coordinate ascent algorithm to (C), updating variables $x$ and $e$ alternatively to optimize the objective. It attempts to scale up (C) with a drastic approximation, running on parts of the image collection to reduce $r$ to only 50 before running on the entire dataset. However, using significantly reduced sets of region proposals hinders its ability to discover multiple objects (Table 1). Moreover, this algorithm is inherently sequential. According to [66], in each iteration of optimizing $x$, an index $i$ is chosen and $x_i$ is updated while $e$ and all $x_j$ with $j \neq i$ are kept fixed. The update of $x_i$ depends on the updated values of other $x_j$ if $x_j$ is updated before $x_i$. This is crucial to guarantee that the objective always increases. If all $x_i$ are updated in parallel, there is no guarantee that the objective would increase. Consequently, this process is not parallelizable, preventing the algorithm from scaling up to datasets with millions of images. We therefore drop the second objective of UOD, and rely only on a fully connected, weighted graph of proposals where edge weights encode proposals' similarity (edge weights can be zeros, see Sec. 3). In turn, we can reformulate UOD as a ranking problem [4, 30, 34, 36, 51], amenable to the panoply of large-scale distributed tools available for eigenvalue problems and link analysis. We consider two different ranking formulations: the first (Q) tackles a quadratic optimization problem, and the second (P) is based on the well-known PageRank algorithm [4, 48]. We combine these two approaches into a joint formulation (LOD) that gives the best results on large-scale datasets. See Sections 3 and 4 for details.

### 2.2 Related work

**Unsupervised object discovery.** Early works on unsupervised object discovery focused on finding groups of images depicting objects of the same categories, employing probabilistic models [58, 61, 69], non-negative matrix factorization (NMF) [62] or clustering techniques [22], see [64] for a survey. In addition to finding image groups, some of these approaches, e.g., multiple instance learning (MIL) [82], graph mining [75], contour matching [39] and topic modeling [58, 61], also output object locations, but focus on smaller datasets with only a handful of distinctive object classes. Saliency detection [80] is related to UOD, but seeks to generate a per-pixel map of saliency scores, while UOD attempts to find bounding boxes around objects in each image without supervision. Object discovery

in large real-world image collections remains challenging due to a high degree of intra-class variation, occlusion, background clutter and appearance of multiple object categories in one image. For this challenging setting, [8] proposes an iterative algorithm which alternates between retrieving image neighbors and localizing salient regions. Based on this approach, [66] is the first to formulate UOD as the optimization problem (C), finding first an approximate solution to a continuous relaxation, then applying block-coordinate ascent to find the solution of the original problem. The first step involves solving a max-flow problem [1] exactly, which is too costly for medium- to large-scale datasets. The datasets have scaled up with successive approaches, from about 3,500 images for [8, 66] to 20,000 for [67], an extension of [66] using a two-stage approach to UOD and a new approach to region proposal design. However, these works are inherently sequential and difficult to scale further. Additionally, [67] operates on reduced sets of region proposals containing only tens of regions, compromising its ability to discover multiple objects (Table 1). In contrast, our proposed approach considers all region proposals, is parallelizable and can be implemented in a distributed way. Thus, it scales well to very large datasets without compromising performance. Indeed, we demonstrate effective UOD in challenging datasets with up to 1.7 million images (Fig. 1).

**Weakly-supervised object localization and image co-localization.** These problems are related to UOD but take advantage of image-level labels. Weakly-supervised object localization (WSOL) considers scenarios where the input dataset contains image-level labels [9]. Most recent WSOL methods localize objects using saliency maps obtained from convolutional features in a neural network [7, 59, 81]. Since networks tend to learn category-discriminating features, various strategies for improving the quality of features for localization have been proposed [2, 10, 76–79]. Co-localization is another line of work where all input images are assumed to contain at least one instance of a single object category. [63] uses discriminative clustering [28] for co-localization in noisy image sets and [29] extends this work to the video setting. [41] learns to co-localize objects by learning sparse confidence distributions, mimicking behaviors of supervised detectors. [70] and [71] observe that activation maps generated from CNN features contain information about object locations. They propose to cluster locations in the images into background and foreground using PCA and return the tight bounding box around the foreground pixels as an object. Since [71] can deal with large-scale datasets and can be easily adapted to UOD, we use it as a baseline in our experiments.

**Ranking applications in computer vision.** The goal of ranking is to assign a global importance rating to each item in a set according to some criterion [48]. Many computer vision problems admit a ranking formulation, including image retrieval [6], object tracking [3], person re-identification [44], video summarization [72], co-segmentation [52] and saliency detection [40]. Several techniques specifically designed for large-scale ranking problems [34, 48] have been used to explore large datasets of images [27] and shapes [16]. PageRank-based approaches in particular have been popular [27, 49, 56] due to their scalability. [31] proposed an algorithm for object discovery that combines appearance and geometric consistency with PageRank-based link analysis for category discovery. However, it does not scale beyond 600 images. A more scalable follow-up work by [32] discovers regions of interest (RoIs) from images with successive applications of PageRank. This algorithm includes two main steps. The first one attempts to find object representatives (*hubs*) from the current RoIs of all images using PageRank. PageRank is then utilized again in the second step to analyze the links between regions in each image and the hubs, this time to update the RoIs of the images. Finally, good RoIs are found by repeating these two steps until convergence. We compare our method to this technique in Sec. 4.1.

## 3 Proposed approach

### 3.1 Quadratic formulation

Let us represent region proposals by a graph $G$ with $N = nr$ nodes, where $n$ is the number of images and $r$ is the number of proposals in each image. Each node corresponds to a proposal, and any two nodes $(p, k)$ and $(q, \ell)$, corresponding to proposals $k$ and $\ell$ of images $p$ and $q$, respectively, are linked by an edge with weight $S_{pq}^{k\ell}$. $G$ is represented by an $N \times N$ symmetric adjacency matrix $W$, consisting of $r \times r$ blocks $S_{pq}$ for $p, q = 1, \ldots n$; $S_{pq}$ is defined in Sec. 2.1 and is computed via PHM algorithm [8] if $p \neq q$, and the diagonal blocks are taken to be zero since only inter-image region similarity matters in our setting. Let $y_i \geq 0$ denote some measure of importance that we want to estimate for node $i$ and set $y = (y_1, \ldots, y_N)^T$. Define the *support* of node $i$ given $y$ as $z_y(i) = \sum_j W_{ij} y_j$ so that taking $z_y = (z_y(1), \ldots, z_y(N))^T$ we have $z_y = Wy$. Intuitively, given $y$, $z_y(i)$ quantifies how well $i$ is connected to (or "supported by") the rest of the nodes $j$ in the graph,

taking into account the similarity $W_{ij}$ between $i$ and $j$ as well as the importance $y_j$ of that node. We would like to find the importance scores that *rank* the nodes as well as possible, so that the order corresponds to their amount of support. As shown by the following lemma, it turns out that this "chicken-and-egg" problem admits a simple solution.

**Lemma 1.** *Suppose $W$ is irreducible (i.e., represents a strongly connected graph $G$). The solution $y^*$ of the quadratic optimization problem:*

$$y^* = \underset{\|t\| \leq 1, t \geq 0}{\operatorname{argmax}} \ t^T W t \tag{Q}$$

*is the unique unit, non-negative eigenvector of $W$ associated with its largest eigenvalue.*

This is a classic result and can be proved using Perron-Frobenius theorem [15, 50]. We include the complete proof in the supplemental material. In our context, $W$ is not, in general, irreducible (i.e., for all pairs $(i, j)$, there exists $m \geq 1$ such that $W^m(i, j) > 0$) since some proposal similarities may be zero. Reminiscent of PageRank [48], we add a small term $\gamma e e^T / N$ to $W$, with $e$ being the vector with all entries equal to 1 in $\mathbb{R}^N$ and $\gamma = 10^{-4}$, deliberately chosen small so that the added term does not influence the similarity score much, to make $W$ irreducible. This term ensures that the resulting ranking is unique and serves the same purpose as the similar term in PageRank. **Note:** since $y^*$ is associated with $W$'s largest eigenvalue $\lambda^*$, which is positive according to the Perron-Frobenius theorem, we have $\lambda^* y^* = W y^* = z_{y^*}$. Hence, the importance score $y_i^*$ of each node is, up to a positive constant, equal to its support, and can thus be used to rank the nodes as desired. Notice that (C) and (Q) are closely related problems when the graph of images in (C) is assumed to be complete. In this case, (C) can be written as $\max_{x \in \{0,1\}^N} x^T W x$, s.t., for all $p$ from 1 to $n$, $\sum_{k=1}^{r} x_{r(p-1)+k} \leq \nu$. Here, we stack $x_i$ $(i = 1, \ldots, n)$ into a vector $x$. (Q) can thus be seen as a continuous relaxation of (C) where the binary variables are replaced by continuous ones, and the linear constraints attaching the proposals to their source images are dropped. The order induced by the dominant eigenvector $y^*$ of $W$ on the nodes of $G$ is reminiscent of the PageRank approach [4, 48] to link analysis. This remark leads to a second approach to UOD through ranking, discussed next.

### 3.2 PageRank formulation

When defining PageRank, [48] does not start from an optimization problem like (Q), but directly formulates ranking as an eigenvalue problem. Following [37], let $A$ denote the transition matrix of the graph associated with a Markov chain, such that $a_{ij} \geq 0$ is the probability of moving from node $j$ to node $i$. In our context, $A$ can be taken as $W D^{-1}$ where $D$ is the diagonal matrix with $D_{jj} = \sum_i W_{ij}$. By definition [4, 48], the PageRank vector $v$ of the matrix $A$ is the unique non-negative eigenvector $v$ of the matrix $P$, associated with its largest (unit) eigenvalue, where $P$ is defined as:

$$P = (1 - \beta)A + \beta u e^T, \tag{P}$$

where $\beta$ is a damping factor. Here, $u$, the so-called personalized vector, is an element of $\mathbb{R}^N$ such that $e^T u = 1$. As noted earlier, the second term ensures that $P$ is irreducible, so that, by the Perron-Frobenius theorem, the eigenvector $v \geq 0$ is unique [38]. The vector $u$ is typically taken equal to $e/N$, but can also be used to "personalize" the ranking by attaching more importance to certain nodes. This leads to the hybrid formulation proposed in the next section. (Q) and (P) are closely related, and the $v$ vector can also be seen as the solution of a quadratic optimization problem [45]. Besides this formal similarity, the goals of the two formulations are also similar. Quoting [48], "a page has a high rank (according to PageRank) if the sum of the ranks of its backlinks is high". The solution of both (Q) and (P), as an eigenvector associated with the largest eigenvalue, provides a ranking based on the support function and can be found with the power iteration algorithm [68]. This algorithm involves only matrix-vector multiplications and can be implemented efficiently in a distributed way.

### 3.3 Using (Q) to personalize PageRank

The above discussion suggests combining the two approaches. We thus propose to use the maximizer of (Q) to generate the personalized vector for (P). (Q) and (P) are two different optimization problems for ranking region proposals, and combining them may help improve the final performance. Intuitively, region proposals with high scores given by (Q) are reliable and we should be able to rank the "objectness" of other regions more accurately based on the "feedback" of these top-scoring proposals. We compute the personalized vector from the solution of (Q) as follows. Given a factor $\alpha$, the top region in each image is chosen as candidates, then the top $\alpha$ percent of regions amongst these candidates are selected. Since only regions that have a high probability of being correct are beneficial, we choose $\alpha$ sufficiently small (see Sec. 4) to select only the most likely correct regions.

Given the set of selected regions, the personalized vector $u$ is the $L_1$-normalized indicator vector with $u_i = 1/K$ where $K$ is the total number of selected regions if proposal $i$ is selected and $u_i = 0$ otherwise. We set the initialization $v_0$ of the power iteration algorithm to $u$ to further bias (P) toward reliable regions found by (Q). In what follows, we refer to this hybrid algorithm as Large-Scale Object Discovery (LOD).

## 4 Experimental analysis

**Datasets.** We consider two large public datasets: C120K, a combination of all images in the training and validation sets of the COCO 2014 dataset [42], except those contain only "crowd" objects, with approximately 120,000 images depicting 80 object classes and OpenImages (Op1.7M) [35], the largest dataset ever evaluated for UOD so far, with 1.7 million images. The latter dataset is 87 times the size of the previous largest dataset evaluated by the state-of-the-art UOD method [67]. We resize all images in this dataset so that their largest side does not exceed 512 pixels. To facilitate ablation studies and comparisons, we also evaluate our methods on C20K, a subset of C120K containing 19,817 images used by [67] and Op50K, a subset of Op1.7M containing 50,000 images.

**Implementation details.** We use the proposal generation method of [67] since it gives the best object discovery performance among the unsupervised region proposal extraction methods [67]. We use VGG16 [60], trained with and without image class labels (Sec. 4.1) on the ImageNet [11] dataset, to both generate (with [67]) and represent (extracting with RoiPool [20]) proposals. We have also experimented with VGG19 [60] and ResNet101 [25], but found they give worse performance, possibly because they are more discriminative and less helpful in localizing entire objects. We compute the similarity score between proposals with the PHM algorithm [8] similar to prior work [8, 66, 67]. For large datasets, computing all score matrices $S_{pq}$ is intractable. In this case, we only compute the similarity scores for the 100 nearest neighbors of each image, computed based on the Euclidean distance between image features from the *fc6* layer. For optimization, we choose $\beta = 10^{-4}$ in (P) and $\alpha = 10\%$ in LOD. To select objects from ranked proposals in an image, we choose proposal $i$ as an object if it has the highest score in the image or the intersection over union (IoU) between $i$ and each of the previously selected object regions is at most $0.3$. When using proposals from [67], which are divided into disjoint groups, we additionally impose that the newly chosen region must be in a group different from the groups of the previously selected objects. See supplemental material for discussions on LOD's sensitivity to hyper-parameters and more implementation details.

**Metrics and evaluation settings.** We consider two settings: single- and the multi-object discovery. In the single-object setting, we return $m = 1$ region per image, which is the region most likely to be an object. In the multi-object setting, we return up to $M$ regions per image, where $M$ is the maximum number of objects in any image in the dataset. Following [43], we assume $M$ is known during evaluation. In a real application, one could use a rough "budget estimate" of the upper bound on how many objects per image one may try to detect. Measuring performance of UOD is always a difficult task due to the ambiguity of the notion of an object in an unsupervised setting: object parts *vs.* objects, individual objects *vs.* crowd objects, *etc*. Following previous works [8, 66, 67], we consider the annotated bounding boxes in the tested datasets as the only correct objects and use them to evaluate our methods. We evaluate UOD results according to the following metrics:

1. *Correct localization score (CorLoc)* – percentage of images correctly localized, i.e., where the IoU score between one of the ground-truth regions and the top predicted region is at least $\sigma = 0.5$. Note that it is equivalent to precision of returned regions. This metric is commonly used to evaluate single-object discovery [8, 66, 67].

2. *Average Precision (AP)* – the area under the precision-recall curve with precision and recall computed at each value of $m$ from 1 to $M$. A ground-truth object is considered discovered if its intersection with any predicted region is at least $\sigma$. This metric is used to evaluate multi-object discovery. We report AP50 where $\sigma = 50$ and AP@[50:95], where we average AP at 10 equally spaced values of $\sigma$ from $0.5$ to $0.95$. Note that AP is different from [67]'s metrics for multi-object discovery, which is the object recall (detection rate) at a predefined value of $m$. This metric depends on the number of selected regions per image $m$ while our metrics do not. In contrast, AP is a standard metric in object detection-like tasks [19–21, 24, 54, 55, 57]. Note that since the precision decreases significantly with increasing $m$, AP appears much smaller than CorLoc.

### 4.1 Large-scale object discovery

In this section, we compare our methods to the state of the art in UOD [32, 67, 71]. We also compare to Edgeboxes (EB) [83], an unsupervised method which outputs regions with an importance score.

EB is a baseline of the type of information bounding boxes alone can provide in our setting. For a fair comparison, we have re-implemented Kim [32] using supervised VGG16 features [60] and proposals from Vo [67]. For Wei [71], we modified their public code, taking bounding boxes around more than one connected component of positive locations from the image *indicator matrix* to return more regions. For other methods, we explicitly evaluate their public code.

**Quantitative evaluation.** We evaluate baselines and the proposed method on C20K, C120K, Op50K and Op1.7M in Table 1. Since state-of-the-art approaches to UOD report results using supervised features [60], we have used these features as well in our comparisons. We additionally report LOD's performance with self-supervised features [18] on these datasets. Overall, LOD obtains state-of-the-art object discovery performance in all settings and datasets. Using VGG16 features [60], it outperforms Kim [32], Wei [71] and EB [83] by large margins: by 26% in single-object discovery and by 14% in multi-object discovery settings. In comparison to Vo [67], LOD performs similarly in the single-object setting, but outperforms Vo [67] by at least 19% in the multi-object setting. This is likely due to the fact that our proposed LOD method considers the full proposal graph and does not reduce the number of region proposals (see supplementary material). It is also noteworthy that LOD scales better than [67] and runs much faster on the large datasets C120K and Op1.7M (Fig. 2). On the Op1.7M dataset, it takes 53.7 hours to run while Vo [67] needs more than a month to finish. It is also interesting that self-supervised features [18] works better with LOD than supervised ones [60], yielding the state of the art performance on Op1.7M dataset.

**Run time.** Next, we compare scalability and run times of the proposed technique and of the baselines. All tested methods ([32, 67, 71, 83] and LOD) use similar pre-processing steps: feature extraction, proposal generation and similarity computation, which are done separately across all images. This is followed in [32, 67] and LOD by an optimization stage. The optimization step in [67] is inherently sequential, but [32] and LOD can be parallelized. In our experiments, we use 4,000 CPUs for preprocessing for all methods, and 48 CPUs for the optimization step in [32] and LOD, the maximum possible with the MatLab parallel toolbox used in our implementation. The timings in Fig. 2 include both pre-processing and optimization, when the latter is used. It can be seen that [32, 71, 83] and LOD scale nearly linearly with the number of images, while [67] exhibits a superlinear pattern. Note that [83] and [71] are 70 times faster than LOD, but at a significant decrease in performance. These methods are not initially designed for object discovery, but serve as good, scalable baselines. Compared to previous top UOD methods, LOD runs at least 2.8 times faster than [32] on all datasets, at least 2 times faster than [67] on datasets between 120K and 1.7 million images. Here, we evaluate only the parallel implementation typical for modern

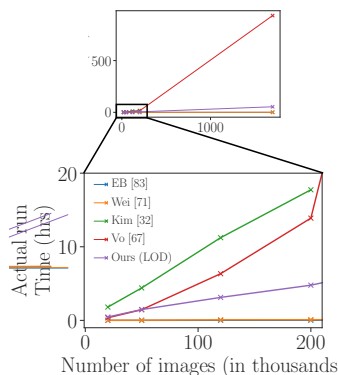

Figure 2: Comparison of run time as a function of input images. LOD achieves significant improvement in performance and/or savings in run time compared to previous works. EB [83] and Wei [71] are linear in the number of images but their run time are very small compared to other methods and look flat in the figure.

computing setups. In a serial implementation, compute times will be similar between top performing UOD methods [32, 67] and LOD, but none of the methods would be able to run on 1.7M images in reasonable time. Note also that additional computational resources can further speed up processing for both Kim [32] and LOD.

**Qualitative evaluation.** We present sample qualitative multi-object discovery results of LOD on C120K and Op1.7M in Fig. 3 (additional Op1.7M results are presented in Fig. 1). LOD discovers both the larger objects (people in the first and sixth images, food items in the second and third images) and the smaller ones (tennis balls and racquet in the first image). It may fail of course, and two typical failure cases are shown on the right of Fig. 3. In the first case, objects are too small and in the second case, LOD returns object parts instead of entire objects. Note that there is some ambiguity in what parts of the image are labelled as ground truth objects. For example, the leaves in the bottom left image are not labelled as objects, while the flowers are.

**Self-supervised features *vs.* supervised features.** LOD and all of the optimization-based baselines [32, 67, 71] rely on a VGG [60]-based classifier trained on ImageNet [11]. In this section, we investigate their performance when the underlying classifier is trained with (Sup) and without (Self) image labels, i.e., in a self-supervised fashion. To obtain self-supervised features, we use a VGG16 model trained with OBoW [18], a recent method which yields state-of-the-art performance in

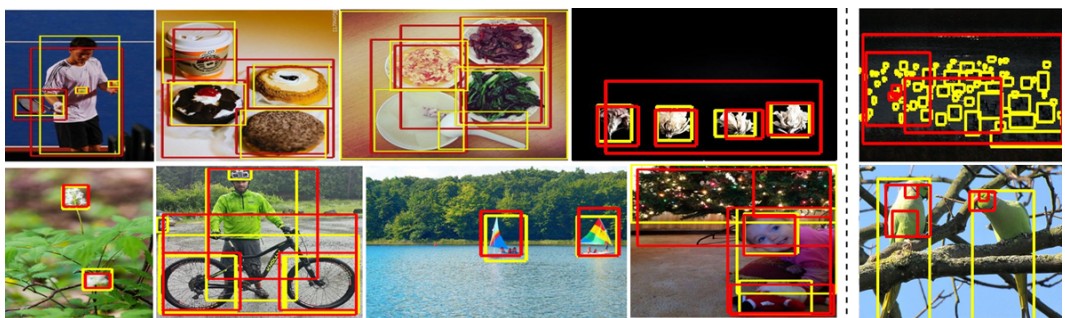

Figure 3: Examples where our method (LOD) succeeds (left) and fails (right) to discover ground-truth objects in the Op1.7M dataset [35]. Ground-truth objects are in yellow, our predictions are in red. Best viewed in color.

object detection after fine-tuning. This model is tested for both the proposal generation and similarity computation steps in optimization-based methods. The results of several variants of each optimization method, depending on the proposal generation algorithm (EB [83], [67]+Self or [67]+Sup) and the region proposal representation (Self or Sup) are presented in Table 2 (left). [67] generates proposals from local maxima of the image's saliency map obtained with CNN features. To evaluate [67]+Self and [67]+Sup for UOD, we assign each proposal a score equal to the saliency of the local maximum it is generated from. If two regions have the same score, the larger one is ranked higher so that entire objects instead of object parts are selected. Finally, when EB [83] proposals are used for [67] and LOD, we multiply their features with their EB scores before computing their similarity.

In general, variants with supervised features perform better in UOD than those with self-supervised features, except for Wei [71] and LOD in single-object discovery on Op50K. Kim [32] is the most dependent on supervised features. Its performance drops by at least 63% when switching to self-supervised features. It is also noteworthy that the performance of Vo [67] and LOD with supervised and self-supervised features on Op50K are much closer than on C20K. This is likely due to the fact that the supervised features [60] are trained on the 1000 ImageNet object classes which contain all of the COCO classes and thus offer a stronger bias toward these classes than the self-supervised features. Using self-supervised features, variants of LOD are the best performer in both single-object discovery (with Vo [67]+Self proposals) and multi-object discovery (with EB proposals). They yield reasonable results on both datasets compared to variants with supervised features. In particular, self-supervised object proposals [67] and self-supervised features, combined with LOD, give the best results of all tested methods on Op50K in single-object discovery. These results show that LOD combined with self-supervised features is a viable option for UOD without any supervision whatsoever.

**Comparing ranking formulations.** We compare the UOD performance of Q, P and LOD with different proposals and features in Table 2 (right). It can be seen that LOD outperforms Q and P in almost all datasets and settings. These results confirm the merit of our proposed method, using Q's solution to personalize PageRank.

### 4.2 Category discovery

Contrary to [8, 66, 67], our work aims specifically at localizing objects in images and omits the discovery of the image graph structure, i.e., identifying image pairs that contain objects of the same category. However, objects localized by our methods can be used to perform this task in a post-processing step. To this end, we define similarity between two images as the maximum similarity between pairs of selected proposals. Similarity is measured using cosine distance between features extracted from *fc6* layer. We compare LOD to [8, 66, 67] in image neighbor retrieval task on VOC_all, a subset of Pascal VOC2007 dataset [13] used as a benchmark in [8, 66, 67]. Similar to these works, we retrieve 10 nearest neighbors per image. Then, CorRet [8] (the average % of retrieved image neighbors that are actual neighbors in the ground-truth image graph over all images) is used to compare different methods. Results are shown in Fig. 4. LOD outperforms [8, 66, 67]. This is surprising since the other methods are specifically formulated to discover image neighbors, while our method is not. This result highlights that our localized objects can be potentially beneficial for other tasks. To go further, we cluster images into categories using proposals selected by our algorithms. Imposing that images are represented by their proposal with the highest score, we perform this task by applying $K$-means on the $L_2$-normalized *fc6* features representing these proposals. We

Table 2: **Left:** UOD performance with supervised [60] (Sup) and self-supervised features [18] (Self) on C20K and Op50K datasets. Region proposals are generated by methods from EB [83] and Vo [67] with different types of features. LOD with self-supervised features yields reasonable results compared to supervised features. Variants of our proposed method LOD yield state-of-the-art performance in all settings. **Right:** A comparison of different ranking methods for UOD. LOD is better than Q and P in most of the cases.

| Opt. | Proposal | Feature | Single-object CorLoc | | Multi-object AP50 | | AP@[50:95] | |
|---|---|---|---|---|---|---|---|---|
| | | | C20K | Op50K | C20K | Op50K | C20K | Op50K |
| None | EB [83] | None | 28.8 | 32.7 | 4.86 | 5.46 | 1.41 | 1.53 |
| | [67]+Self | None | 29.7 | 39.8 | 2.47 | 3.72 | 0.61 | 1.0 |
| | [67]+Sup | | 23.6 | 38.1 | 4.07 | 4.81 | 1.03 | 1.39 |
| Wei [71] | None | Self | 37.9 | 42.4 | 2.53 | 3.13 | 0.69 | 0.9 |
| | | Sup | 38.2 | 34.8 | 2.41 | 1.86 | 0.73 | 0.6 |
| Kim [32] | EB [83] | Self | 5.5 | 5.4 | 0.64 | 0.79 | 0.13 | 0.15 |
| | | Sup | 15.6 | 20.2 | 1.96 | 2.56 | 0.36 | 0.47 |
| | [67]+Self | Self | 4.7 | 4.6 | 0.13 | 0.29 | 0.02 | 0.05 |
| | [67]+Sup | Sup | 35.1 | 37.0 | 3.93 | 4.13 | 0.96 | 0.98 |
| Vo [67] | EB [83] | Self | 35.6 | 43.6 | 3.34 | 4.43 | 0.99 | 1.39 |
| | | Sup | 40.2 | 44.0 | 4.0 | 4.47 | 1.21 | 1.41 |
| | [67]+Self | Self | 37.8 | 48.1 | 2.65 | 4.19 | 0.82 | 1.45 |
| | [67]+Sup | Sup | **48.5** | 48.0 | 5.18 | 4.98 | 1.62 | 1.58 |
| LOD | EB [83] | Self | 35.5 | 39.7 | 5.87 | 6.73 | 1.57 | 1.76 |
| | | Sup | 38.9 | 41.3 | 6.52 | 7.01 | 1.76 | 1.86 |
| | [67]+Self | Self | 41.1 | **49.5** | 4.56 | 6.37 | 1.29 | 1.87 |
| | [67]+Sup | Sup | **48.5** | 48.1 | **6.63** | 6.46 | **1.98** | **1.88** |

| Opt. | Proposal | Feature | Single-object CorLoc | | Multi-object AP50 | | AP@[50:95] | |
|---|---|---|---|---|---|---|---|---|
| | | | C20K | Op50K | C20K | Op50K | C20K | Op50K |
| Q | EB [83] | Self | 32.8 | 40.3 | 4.15 | 6.43 | 1.07 | 1.67 |
| | | Sup | 36.0 | 41.1 | 5.72 | 6.49 | 1.47 | 1.7 |
| | [67]+Self | Self | 38.7 | 48.9 | 4.38 | 6.39 | 1.17 | 1.84 |
| | [67]+Sup | Sup | 43.8 | 47.5 | 6.21 | 6.66 | 1.74 | **1.88** |
| P | EB [83] | Self | 35.5 | 39.7 | 4.91 | 6.73 | 1.34 | 1.75 |
| | | Sup | 38.9 | 41.3 | 6.51 | 6.99 | 1.76 | 1.86 |
| | [67]+Self | Self | 41.2 | 49.5 | 4.38 | 6.13 | 1.24 | 1.81 |
| | [67]+Sup | Sup | 47.5 | 47.8 | 6.25 | 6.19 | 1.87 | 1.81 |
| LOD | EB [83] | Self | 35.5 | 39.7 | 5.87 | 6.73 | 1.57 | 1.76 |
| | | Sup | 38.9 | 41.3 | 6.52 | 7.01 | 1.76 | 1.86 |
| | [67]+Self | Self | 41.1 | 49.5 | 4.56 | 6.37 | 1.29 | 1.87 |
| | [67]+Sup | Sup | 48.5 | 48.1 | **6.63** | 6.46 | **1.98** | **1.88** |

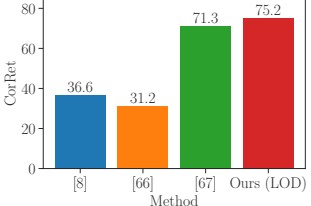

Figure 4: Comparison to prior work on the image neighbor retrieval task (CorRet, ↑).

Table 3: Purity (↑) of our clustering method compared to the state of the art in category discovery on the SIVAL dataset [53]. Following prior work, we perform the task on a partition of the dataset and report the average purity on its parts as the final result. Results of other methods are from [75].

| Dataset/Method | Ours (LOD) | [75] | [82] | [14] | [73] | [74] | [31] | [33] |
|---|---|---|---|---|---|---|---|---|
| SIVAL1 | **97.4** | 89.0 | 95.3 | 80.4 | 39.3 | 38.0 | 27.0 | 45.0 |
| SIVAL2 | **99.0** | 93.2 | 84.0 | 71.7 | 40.0 | 33.3 | 35.3 | 33.3 |
| SIVAL3 | 88.3 | 88.4 | 74.7 | 62.7 | 37.3 | 38.7 | 26.7 | 41.3 |
| SIVAL4 | **97.7** | 87.8 | 94.0 | 86.0 | 33.0 | 37.7 | 27.3 | 53.0 |
| SIVAL5 | **94.3** | 92.7 | 75.3 | 70.3 | 35.3 | 37.7 | 25.0 | 48.3 |
| Average | **95.3** | 90.2 | 84.7 | 74.2 | 37.0 | 37.1 | 28.3 | 44.2 |

conduct experiments on the SIVAL [53] dataset, a popular benchmark for this task. This dataset consists of 25 object categories, each containing about 60 images. Following [82], we partition the 25 object classes into 5 groups, named SIVAL1 to SIVAL5, and use purity (average percentage of the dominant class in the clusters) as an evaluation metric. Intuitively, purity measures the extent to which a cluster contains images of a single dominant class. A comparison between our method and other popular object category discovery methods is given in Table 3. It can be seen that our method outperforms the state of the art by a significant margin, attaining an average purity of 95.3. It is also noteworthy that the performance drops to 23.7 when the features of entire images are used instead of the representative top proposals. This finding shows that our performance gain is in great part due to the object localization performance of our method. Since individual images in the SIVAL dataset [53] contain only one object, we conduct a similar experiment on the more challenging VOC_all [13] dataset. In this experiment, a histogram is computed for each cluster, showing the score of each ground-truth object category (a category score is the sum of contributions of all its images). An image contribution is computed as $1/nc$, with $c$ is the number of object categories appearing in the image and $n$ is the number of images in the cluster. We then match the clusters to the ground-truth categories by solving a stable marriage problem with the Gale–Shapley algorithm [17] using the preference orders induced by the histograms.

The confusion matrix generated by combining these histograms, revealing the correspondence between the clusters and the classes, is shown in Fig. 5. Our method is able to discover 17 categories (which are dominant in at least one cluster) out of 20 ground-truth categories. As for the three undiscovered categories: *sheep* is dominated by similar class *cow* in cluster 10; *sofa* is dominated by co-occurring class *chair* in cluster 9; *dinningtable* suffers from being often largely occluded in images. Interestingly, it seems that our method might be used to discover pairs of categories that often appear together, for instance: *bicycle* and *person*, *horse* and *person*, *motorbike* and *person* (clusters 2, 13 and 14 have two corresponding dominating classes each). Quantitatively, using the top

extracted proposals from our method achieves a purity of 68.6 on this dataset, which is better than the purity of 61.8 obtained when features of entire images are used.

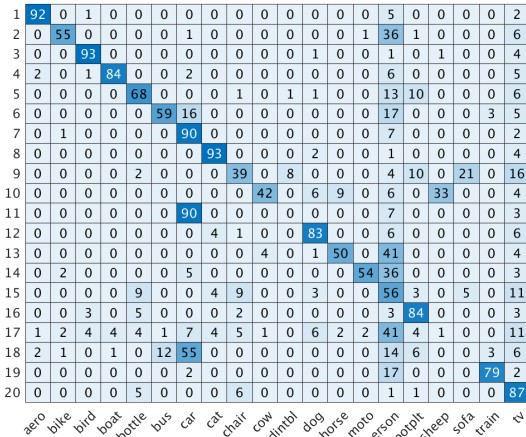

Figure 5: Confusion matrix revealing links between the object classes and the clusters found by (LOD) on VOC_all.

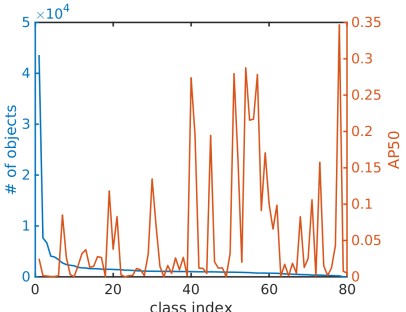

Figure 6: Performance of LOD by object category and category frequency (number of object occurrences of each category) on the C20K dataset. Results are reported with Average Precision at $\sigma = 0.5$ (AP50), higher numbers are better. Object categories are indexed in decreasing order of category frequency. Performance of LOD is not well-correlated (correlation $-0.09$) with category frequency.

### 4.3 Discussions

Without a formal definition of objects, casting objects as frequently appearing salient visual patterns is natural. However, findings could be biased toward popular object classes and ignore rare classes in image collections that contain a long-tail distribution of object classes. To have an insight to this potential bias, we compute LOD's performance by object category on C20K dataset. Surprisingly, we have observed little correlation between the performance on an object class and its appearance frequency (the corresponding correlation is only $-0.09$, see Figure 6). A possible explanation is that even though we rank all regions in the image collection at once, we choose objects (based on the ranking) on the image level. Therefore, regions can be selected as objects if they stand out more from the background and are better connected in the graph than other regions in the same image, even if they represent objects of a rare class.

## 5 Conclusion and future work

We have demonstrated a novel formulation of unsupervised object discovery (UOD) as a ranking problem, allowing application of efficient and distributed algorithms used for link analysis and ranking problems. In particular, we have shown how to apply the personalized PageRank algorithm to derive a solution, and proposed a new technique based on eigenvector computation to identify the personalized vector in Pagerank. The proposed LOD algorithm naturally admits a distributed implementation and allows us to scale up UOD to the OpenImages [35] dataset (Op1.7M) with 1.7M images, 87 larger than datasets considered in the previous state-of-the-art technique [67], and outperforms (in single- and multi-object discovery) all existing algorithms capable of scaling to this size. In multi-object discovery, LOD is better than all other methods on medium and large-scale datasets. State-of-the-art solutions to UOD rely on supervised region proposals [8] or features [71, 67], thus their output requires at least in part on some sort of supervision. We have proposed to combine LOD with self-supervised features, offering a solution to fully unsupervised object discovery. Finally, we have shown that LOD yields state-of-the-art results in category discovery which is obtained as a post-processing step. A limitation of our method is that it works well only with VGG-based features which prevents it from benefiting from more powerful features [25, 26]. Future work will be dedicated to investigating these features for LOD. Another interesting avenue for future research is to better discover smaller objects.

Effective solutions to UOD have the potential for a great impact on existing visual classification, detection, and interpretation technology by harnessing the vast amounts of non-annotated image data available on the Internet today. In turn, the application of this technology has its known potential benefits (from natural human computer interfaces to X-ray image screening in medicine) and risks (from state-sponsored surveillance or military target acquisition). We believe that such concerns are ubiquitous in machine learning in general and computer vision in particular, and beyond the scope of this scientific presentation.

## Acknowledgments and Disclosure of Funding

This work was supported in part by the Inria/NYU collaboration, the Louis Vuitton/ENS chair on artificial intelligence and the French government under management of Agence Nationale de la Recherche as part of the "Investissements d'avenir" program, reference ANR19-P3IA-0001 (PRAIRIE 3IA Institute). Elena Sizikova was supported by the Moore-Sloan Data Science Environment initiative (funded by the Alfred P. Sloan Foundation and the Gordon and Betty Moore Foundation) through the NYU Center for Data Science. Huy V. Vo was supported in part by a Valeo/Prairie CIFRE PhD Fellowship. We thank Spyros Gidaris for providing the VGG16-based OBoW model. Finally, we thank anonymous reviewers for their helpful suggestions and feedback for the paper.

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
