# Large-Scale Unsupervised Object Discovery
# – Supplementary Material –

**Huy V. Vo**[1,2]     **Elena Sizikova**[3]     **Cordelia Schmid**[1]     **Patrick Pérez**[2]     **Jean Ponce**[1,3]

[1]INRIA, Département d'informatique de l'ENS, ENS, CNRS, PSL University, Paris, France
[2]Valeo.ai     [3]Center for Data Science, New York University
{van-huy.vo, cordelia.schmid, jean.ponce}@inria.fr
es5223@nyu.edu    patrick.perez@valeo.com

## A   Appendix

We include in this appendix additional information about the proposed method, including implementation details, experimental results, visualizations and proofs.

### A.1   Additional implementation details

**PHM algorithm.**  We use the probabilistic Hough matching (PHM) algorithm (1) to compute region similarity scores in our implementation due to its effectiveness in object discovery (1; 10; 11). Given two images $i$ and $j$, the PHM algorithm (1) computes the match between region $k$ of image $i$ and region $l$ of image $j$ a score defined as:

$$S_{ij}^{kl} = a_{ij}^{kl} \sum_{k',l'} K_{ij}^{kl,k'l'} a_{ij}^{k'l'}, \tag{PHM}$$

where $a_{ij}^{kl}$ is the appearance similarity between the two regions and $K_{ij}^{kl,k'l'}$ measures how compatible two potential matches $(k,l)$ and $(k',l')$ are geometrically. The "Hough" part of the algorithm's name comes from the associated geometric voting procedure. In our implementation, $a_{ij}^{kl}$ is the dot product of the unnormalized CNN features associated with the two regions, and $K_{ij}^{kl,k'l'}$ is computed by comparing the matches $(k,l)$ and $(k',l')$ against a set of discretized geometric transformations. See (1; 10) for more details.

**Parallel power iterations.**  We solve Q, P, and LOD with the power iteration method (12) (Algorithm 1 below). Since the adjacency matrix (in Q) and the PageRank matrix (in P) are very large, we divide them into chunks of consecutive rows of approximately equal size. At iteration $t$ in the optimization, these chunks are loaded in parallel into multiple processors' memories for multiplication with the current iterate $x_t$. The results of these operations are chunks of the new vector $x_{t+1}$ which is then assembled from them. We run up to $T = 50$ iterations of the power method in each experiment.

### A.2   Influence of hyper-parameters

The proposed method has two important hyper-parameters, the damping factor $\beta$ in PageRank and the scalar $\alpha$ used to select reliable object candidates in LOD. In practice, $\beta$ should be small so as not to change much the weight matrix $A$ and $\alpha$ should also be small since we only want to select a few top-scoring proposals. We have evaluated PageRank for object discovery on C20K and Op50K datasets[1] with increasing values of $\beta$, ranging from $10^{-5}$ to $10^{-1}$, and present the results in Table 1 (left).

---

[1]We remind that C20K is a subset of COCO (7) (C120K) and that Op50K is a subset of OpenImages (6) (Op1.7M).

35th Conference on Neural Information Processing Systems (NeurIPS 2021).

**Algorithm 1:** Parallel power iterations for finding the first eigenvector of a matrix $A$.

---

**Result:** The first eigenvector of $A$.

**Input:** Number $M$ of matrix chunks, chunks of matrix rows $A_1,\ldots,A_M$ of $A$, number of region proposals $N$, norm $L_p$ ($p = 1$ for P and $p = 2$ for Q), number of iterations $T$.

**Initialization:** $x_0 = \dfrac{1}{\|e_N\|_p} e_N$.

**for** $t = 0$ *to* $T - 1$ **do**
$\quad$ In parallel in multiple processors:
$\quad$ **for** $i = 1$ *to* $M$ **do**
$\quad\quad$ Load matrix chunk $A_i$ into memory.
$\quad\quad$ Compute the $i$-th chunk $x_{t+1,i} = A_i x_t$.
$\quad$ **end**
$\quad$ In the main processor, assemble $x_{t+1}$ from its chunks: $x_{t+1} = [x_{t+1,1}; x_{t+1,2}; \ldots; x_{t+1,M}]$.
$\quad$ Normalize: $x_{t+1} = \dfrac{1}{\|x_{t+1}\|_p} x_{t+1}$.
**end**
Return $x_T$.

---

Table 1: Influence of the damping factor $\beta$ on PageRank's performance (left) and of the selection factor $\alpha$ on LOD's performance (right) on the C20K and Op50K datasets.

| $\beta$ | Single-object | | Multi-object | | | |
|---|---|---|---|---|---|---|
| | CorLoc | | AP50 | | AP@[50:95] | |
| | C20K | Op50K | C20K | Op50K | C20K | Op50K |
| $10^{-5}$ | 48.0 | 47.8 | 6.3 | 6.13 | 1.89 | 1.8 |
| $10^{-4}$ | 48.0 | 47.8 | 6.29 | 6.19 | 1.89 | 1.81 |
| $10^{-3}$ | 47.9 | 47.7 | 6.22 | 6.08 | 1.87 | 1.78 |
| $10^{-2}$ | 47.0 | 47.0 | 5.82 | 5.69 | 1.76 | 1.68 |
| $10^{-1}$ | 40.0 | 38.8 | 4.45 | 4.14 | 1.34 | 1.22 |

| $\alpha$ | Single-object | | Multi-object | | | |
|---|---|---|---|---|---|---|
| | CorLoc | | AP50 | | AP@[50:95] | |
| | C20K | Op50K | C20K | Op50K | C20K | Op50K |
| 0.05 | 48.4 | 48.2 | 6.63 | 6.5 | 1.99 | 1.89 |
| 0.10 | 48.5 | 48.1 | 6.63 | 6.46 | 1.98 | 1.88 |
| 0.15 | 48.5 | 48.2 | 6.64 | 6.49 | 1.99 | 1.89 |
| 0.20 | 48.5 | 48.2 | 6.64 | 6.48 | 1.99 | 1.89 |

This experiment shows that the performance of PageRank begins to drop when $\beta$ becomes larger than $10^{-3}$ and deteriorates significantly when it exceeds $10^{-2}$. It does not depend much on $\beta$ when this parameter is small enough (less than $10^{-3}$). We choose $\beta = 10^{-4}$ in our implementation. We have also evaluated LOD with different values of $\alpha$, taken in $\{0.05, 0.1, 0.15, 0.2\}$, which amounts to selecting 5%, 10%, 15% and 20% of candidates respectively, and show the results in Table 1 (right). As long as $\alpha$ is reasonably small, its value does not significantly affect the performance of LOD. We choose $\alpha = 0.1$ in our implementation.

We have also assessed on the C20K and Op50K datasets the sensitivity of LOD to $N$, the number of initial image neighbors, and to $\gamma$, the parameter controlling the strength of the small perturbation added to the score matrix $W$. For $N$, we have tried different values from 100 to 500 and found that the performance improves only slightly when more neighbors are considered. However, the computational cost increases linearly with $N$ and we find that using 100 neighbors is a good compromise for our datasets. As discussed later, the number of neighbors might need to be changed in case of (undetected) near duplicates or in the case of videos (with successive, highly similar frames). For $\gamma$, we have varied its value in $\{10^{-6}, 10^{-5}, 10^{-4}, 10^{-3}, 10^{-2}\}$ and found that the performance does not vary much ($\leq 0.5\%$) when $\gamma \leq 10^{-4}$ and slightly degrades when $\gamma \geq 10^{-3}$. This shows that LOD is insensitive to $\gamma$ as long as it is small enough.

### A.3 Influence of the number of object proposals on object discovery performance

Unlike (11), we are able to use almost all the regions produced by the proposal algorithm (2000 regions per image at most) thanks to the good scalability of our formulation. On average, we have 814 and 850 regions per image on C20K and Op50K, respectively. We have evaluated LOD on C20K and Op50K using different numbers of proposals (see Table 2) and observed that its performance improves with additional region proposals, notably in the multi-object setting. This observation partly

explains our better performance compared to (11) (which places a limit on the number of regions for computational reasons) and the benefit of using all region proposals.

Table 2: Influence of the number of object proposals on performance of LOD.

| # of regions | C20K | | | Op50K | | |
|---|---|---|---|---|---|---|
| | CorLoc | AP50 | AP@[50:95] | CorLoc | AP50 | AP@[50:95] |
| 50 | 40.9 | 4.5 | 1.22 | 42.0 | 4.55 | 1.31 |
| 100 | 44.0 | 5.38 | 1.47 | 43.4 | 5.1 | 1.4 |
| 200 | 46.5 | 6.13 | 1.71 | 45.6 | 5.83 | 1.61 |
| 400 | 48.0 | 6.6 | 1.91 | 47.1 | 6.32 | 1.77 |
| All | 48.5 | 6.63 | 1.98 | 48.1 | 6.46 | 1.88 |

## A.4   Influence of underlying features

We use features from a VGG16 (9) model trained for image classification on ImageNet (2) in our main experiments. We have also tested LOD with features from VGG19 (9) and ResNet50 (4) and present the results on C20K and Op50K in Table 3. Although VGG19 and ResNet50 give better results in image classification (4; 9), they perform worse than VGG16 in object discovery with LOD. This may be due to the fact that they are more discriminative, focusing mostly on the most prominent object parts thus less helpful in localizing entire objects, although we do not have a definitive answer (yet) for this.

Table 3: LOD performance with VGG19 (9), ResNet50 (4) and VGG16 (9) features on C20K and Op50K datasets. Although the latter are more powerful in image classification, VGG16 features yield the best results in object discovery with LOD.

| Features | Single-object | | Multi-object | | | |
|---|---|---|---|---|---|---|
| | CorLoc | | AP50 | | AP@[50-95 | |
| | C20K | Op50K | C20K | Op50K | C20K | Op50K |
| VGG19 (9) | 47.4 | 45.1 | 6.27 | 5.57 | 1.84 | 1.58 |
| ResNet50 (4) | 35.4 | 45.9 | 4.08 | 5.59 | 1.05 | 1.46 |
| VGG16 (9) | **48.5** | **48.1** | **6.63** | **6.46** | **1.98** | **1.88** |

## A.5   Multi-object discovery performance according to a detection rate metric

Contrary to (11), we have evaluated multi-object discovery performance using average precision (AP) instead of detection rate (11) (DetRate), which can also be thought of recall over ground-truth objects. We argue (in the main body of our submission) that plain DetRate is not a good metric for multi-object discovery since it depends on the number $m$ of regions returned per image, which is pre-defined. Beside the fact that there is *a priori* no optimal choice for $m$, evaluating the performance at a single value of $m$ does not capture the range of possible performances. AP, on the other hand, summarizes the performance at different values of $m$.

Despite these remarks, we present here for completeness the multi-object discovery performance in DetRate for LOD and the baselines in Table 4. In addition to computing DetRate at $m = 5$ as in (11), we also consider $m = \bar{m}$ where $\bar{m}$ is the average number of ground-truth objects per image in the dataset, which is 7 for C20K and C120K, and 8 for Op50K and Op1.7M. The results show that LOD significantly outperforms the baselines in all datasets when detection rate is computed at $m = 5$. It also performs better than the others when detection rate is computed at $m = \bar{m}$, except for Edgeboxes (EB) (14) on Op1.7M dataset. However, we stress again that we think detection rate is not a natural metric for multi-object discovery. We show in the main paper that LOD is significantly better than all baselines in all datasets according to AP, which we think is a more appropriate metric for object discovery.

## A.6   More qualitative results

We show additional examples on COCO (7) in Fig. 1 and on OpenImages (6) in Fig. 2 for which LOD successfully discovers objects. We also present some failure cases in Fig. 3. LOD typically

Table 4: Large-scale multi-object discovery performance and comparison to the state of the art on COCO (7), OpenImages (6) and their respective subsets C20K and Op50K, as measured by detection rate

| Method | Multi-object | | | | | | | |
| | DetRate ($m = 5$) | | | | DetRate ($m = \bar{m}$) | | | |
| | C20K | C120K | Op50K | Op1.7M | C20K | C120K | Op50K | Op1.7M |
|---|---|---|---|---|---|---|---|---|
| EB (14) | 12.0 | 12.1 | 12.5 | 12.5 | 14.5 | 14.5 | 16.0 | **16.0** |
| Wei (13) | 6.8 | 6.9 | 5.7 | 5.7 | 6.8 | 6.9 | 5.7 | 5.7 |
| Kim (5) | 10.5 | 10.6 | 10.8 | - | 12.1 | 12.2 | 12.9 | - |
| Vo (11) | 12.3 | 11.8 | 11.8 | - | 13.3 | 12.7 | 13.1 | - |
| Ours (LOD) | **14.2** | **14.2** | **14.0** | **13.7** | **15.7** | **15.7** | **16.2** | 15.8 |

fails to discover objects that are too small (images 1 to 5) or only discovers the most discriminative object parts instead of entire objects (images 6 to 8). In some cases, LOD discovers objects that are not annotated: entrance in image 1, tower in image 2 and flower branch in image 4.

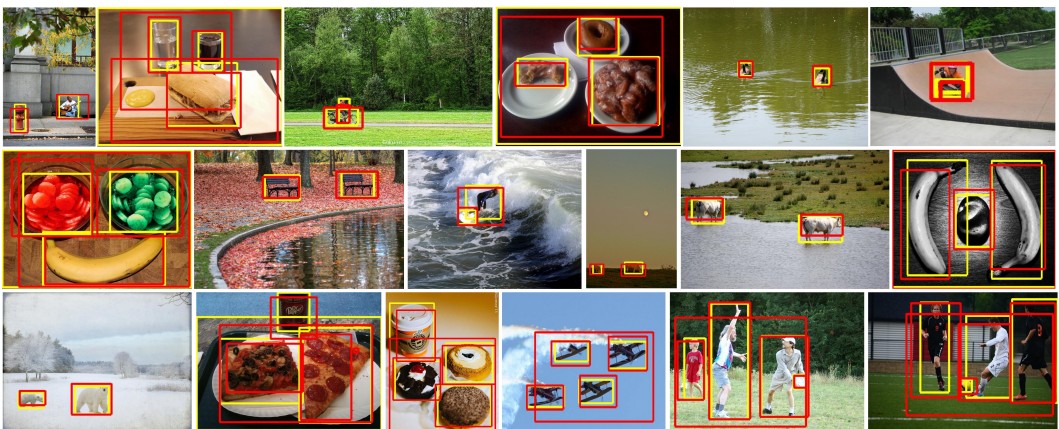

Figure 1: Examples in the COCO (7) dataset where LOD successfully discovers ground-truth objects. Ground-truth boxes are in yellow and our predictions are in red.

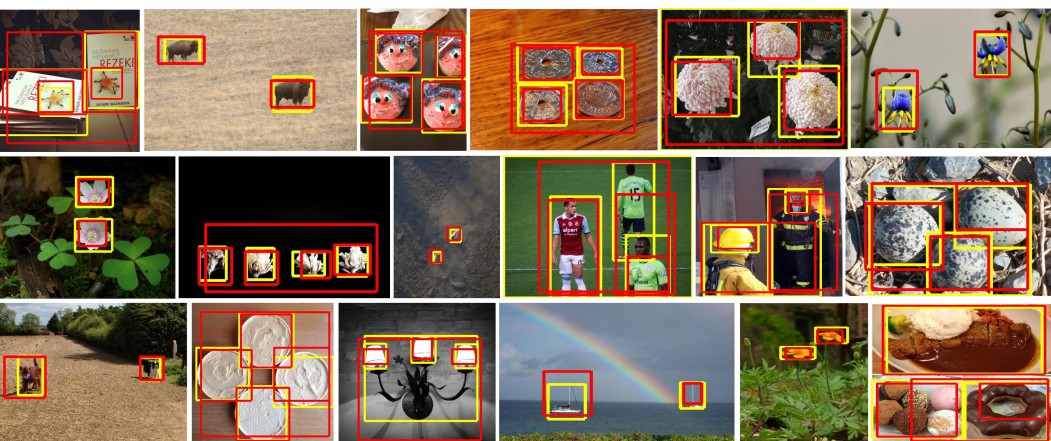

Figure 2: Examples in the OpenImages (6) dataset where LOD successfully discovers ground-truth objects. Ground-truth boxes are in yellow and our predictions are in red.

## A.7 Proof of Lemma 1.

*Proof.* Since $W$ is symmetric, all its eigenvalues are real and it can be diagonalized by an orthonormal basis of its eigenvectors. The maximizer of $t^T W t$ in the unit ball is the unit eigenvector of $W$ associated with its largest eigenvalue $\lambda^*$. Given that $W$ is irreducible, it has a unique, unit,

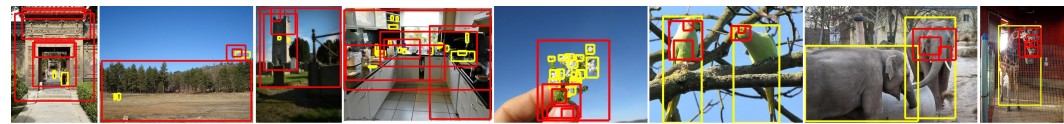

Figure 3: Examples in the COCO (7) and OpenImages (6) datasets where LOD fails to discover ground-truth objects. Ground-truth boxes are in yellow and our predictions are in red.

non-negative eigenvector associated with its largest eigenvalue, according to the Perron-Frobenius theorem (3; 8). ∎

**Note:** This is a classic result, only included here for completeness.