# OpenReview forum: "Large-Scale Unsupervised Object Discovery"
_NeurIPS.cc/2021/Conference — NeurIPS 2021 Poster_

### Official Review · Reviewer_FJwA · 2021-07-02

**Rating:** 6
**Confidence:** 3

**Summary:**

This paper introduces an efficient approach for Unsupaervised Object Discovery (UOD). This efficiency was achieved by considering the exising UOD formulation as a ranking problem by modifying the binary importance vector as having a floating value. This modification allows to apply this approach to a very large-scale dataset (up to 1.7M images) without compromising accuracy.

**Ethical Concerns:**

I did not find any ethical concerns in this paper.

**Limitations And Societal Impact:**

As I mentioned, my main concern is that its technical contribution is very limited. Other than that, the impact of this work on large-scale object discovery is quite satisfactory.

**Main Review:**

- pros
1. It was very nice to reformulate the existing UOD formulation to be fit to a PageRank solution.
2. First to introduce an UOD approach to be applied in the large-scale dataset (Op1.7M).

- cons
1. Reformulation that modifying the binary importance vector y as having a float value to become PageRank vector v is incremental.

2. A more explanation is needed on how this method can be processed in parallel and why other methods can not. P matrix computed on the large-scale dataset must be very large. How can this matrix be divided into multiple chunks?

In L112, what does it mean "inherently sequential" for other baselines?


---------------------------------------------------------------------------------------------
After reading authors' responses and other reveiwers' comments, I raise my rating to "lean to accept". My concerns are well addressed.

**Time Spent Reviewing:**

24 hours

---

> ### Author Response · Authors · 2021-08-10
> **We thank the reviewer for the helpful comments and suggestions and address the reviewer's concerns below.**
>
> **Q: Reformulation that modifying the binary importance vector y as having a float value to become PageRank vector v is incremental**
>
> Although we show that our formulation and the formulation in [66,67] are related, there is a fundamental difference between the two. [66,67] look at the graph of images, where the relation between images is inferred from their proposals. In their case, object discovery is formulated as **a discrete optimization problem**. On the other hand, we consider a weighted graph of proposals across all images (no image constraint) and solve **an eigenvector problem** on its weight matrix to find objects. This change proves to be valuable as it allows us to scale up to 1.7M images (87 times larger than [67]) and outperform the state-of-the-art methods on medium-size datasets by up to 32% in Average Precision (AP).
>
> **Q: A more explanation is needed on how this method can be processed in parallel and why other methods can not. P matrix computed on the large-scale dataset must be very large. How can this matrix be divided into multiple chunks?**
>
> Our method consists of three steps: proposal generation, feature extraction and optimization. Since the first two steps are done separately on each image, they can be easily parallelized. The key innovation that we propose is the optimization step, which is formulated as finding the leading eigenvector of a big matrix. We use a simple implementation of the power iteration algorithm [68] which consists only of matrix-vector multiplications and can be parallelized. We describe it in Algorithm 1 in the supplemental material. We present briefly here how we build chunks of matrix $W$. The matrix $P$ used in PageRank is derived from $W$ and is processed similarly. As presented in Section 3.1, $W$ is an $n \times n$ block matrix where the block at row $p$ and column $q$ contains the similarity $S_{pq}$ between regions in images $p$ and $q$. The $i$-th chunk $W_i$ of $W$ consists of $l_i$ consecutive rows of blocks from $bb_i$ to $eb_i$ (begin block  and end block) which correspond to consecutive rows of $W$ with indices from $br_i$ to $er_i$ (begin row and end row), respectively. As a result, $W_i$ has size $(er_i-br_i+1) \times N$ where $N$ is the total number of regions in the image collection. To build $W_i$, we simply compute in parallel the similarity $S_{pq}$ with $bb_i \leq p \leq eb_i$ and assemble them into their position in $W_i$. We explain why [66,67] cannot be parallelized below.
>
> **Q: In L112, what does it mean "inherently sequential" for other baselines?**
>
> [66,67] formulate object discovery as a discrete optimization problem (C) and solve it by block coordinate ascent, iteratively alternating between the optimization of variables $x$ and $e$, while the variables $x_i$ are optimized sequentially. The precise steps of this alternative algorithm are detailed in Section 2.5 in [66].  Concretely, in each iteration of optimizing $x$, an index $i$ is chosen and $x_i$ is updated while $e$ and all $x_j$ with $j \neq i$ are kept fixed. The update of $x_i$ depends on the updated values of other $x_j$ if $x_j$ is updated before $x_i$. This is crucial to guarantee that the objective always increases. If all $x_i$ are updated in parallel, there is no guarantee that the objective would increase. This process therefore cannot be parallelized and the optimization of [66,67] is sequential by nature. These points will be clarified.

---

### Official Review · Reviewer_1AuG · 2021-07-15

**Rating:** 7
**Confidence:** 4

**Summary:**

This is a very interesting paper. It builds upon object discovery formulation by [Vo et al., CVPR’19; ECCV’20], which formulates the discovery as (co-)detection of frequently occurring patterns within an image collection. This approach assumes input in the form of object proposals (together with a feature extractor or related mechanism that will be used to determine pairwise similarities) and formalizes object co-detection as a combinatorial optimization problem that seeks object proposals with high connectivity across the dataset. It has been established by [Vo et al., CVPR’19; ECCV’20] that this approach performs notably better compared to sliding window-based approaches that rank proposals in images independently [e.g., Zitnick et al., ECCV’14]. The paper makes a case that optimization methods proposed in prior work, do not scale to very large datasets, largely because they are not parallelizable. To tackle the scalability issue, this paper draws parallels between the discovery objective [Vo et al., CVPR’19] and object ranking, cast as a quadratic optimization problem, and well-known PageRank algorithm, as well as a combination of the two. Under this formulation, optimal ranking of proposals can be obtained via Eigenvalue decomposition of the adjacency matrix using the power iteration algorithm. While single-core implementation does not present a significant advantage over solvers proposed to tackle the original optimization formulation [Vo et al., CVPR’19; ECCV’20], this approach is highly parallelizable and, in turn, allows to apply this method to very large datasets, such as OpenImages, containing over 1.7M images and is on-par or slightly better in terms of the object discovery performance.

**Ethical Concerns:**

I see no ethical issues.

**Limitations And Societal Impact:**

As I noted above, authors should address limitations better.

**Main Review:**

Pros

———————

* This paper is very well written, motivated, and presents a very good case. Object discovery in image collection is indeed a very relevant problem, as we are slowly approaching the limits of the supervised learning paradigm. Nowadays, image data is abundant; therefore, I see the contribution of methods that can scale well to very large datasets as very valuable to our community.
* In general, I find that the technical contribution of this paper stands on very solid ground: I love the insight that the formulation by [Vo et al., CVPR’19] can be re-interpreted as a ranking problem over a fully-connected graph of proposals and establishing the link to well-understood and mature methodology for link analysis.
* The implications of this insight are important and will allow performing object discovery on (abundant) large-scale unlabelled photo collections. The contribution is orthogonal to (from my side anticipated) future efforts in unsupervised object-centric representation learning.
* The paper presents solid experimental analysis on COCO dataset (and subsets), as well as the large-scale OpenImage dataset. Analysis shows that the only competitive method [Vo et al., ECCV’20] is not applicable to OpenImages (I do have some criticism on this point, though- see the weaknesses section). I also appreciate the detailed discussion on the impact of different object proposals and features (ImageNet-supervised and self-supervised), and the comparison between three variants of this ranking formulation.
* Besides parallelization, it is also very nice that this formulation is scalable enough that it doesn’t need to limit the number of object proposals per image, which I assume is the main reason behind slightly better performance compared to [Vo et al., ECCV’20]. At the same time, I would also like to note that it would be nice to discuss this experimentally.


Cons

———————

* The most problematic aspect of this paper is the very vague definition of what “other top-performing methods cannot handle” (Tab 1.). This is further related to Sec. 4.1, run time discussion, which states that “[32] and [67] did not finish in reasonable time”. What is considered to be a reasonable time? One week (par. “Quantitative evaluation” mentions that)? For example, training neural networks can easily take one week or more, and if a method needs more than a week (2 weeks?) to finish, I would not say that it is not applicable to such a large dataset. I understand that one cannot wait for the optimization to converge for months, but I am really in the dark here what kind of time spans we are talking about, and I would like to see a discussion on this (also the final version of the paper must be more precise here). Also, let me remark here that “reasonable time” will be different in the future with better compute (although it is true that datasets will be growing as well).
* It can be seen that optimization-based methods scale linearly (i.e., scale very well). Therefore we can also expect that these methods should be very applicable in the future. I definitely see that the major advantage of the proposed method due to the fact that it is nicely parallelizable, but this aspect is related to the core contribution of this paper. Finally, I do not understand why EB runtime is constant? It should be linear wrt. the number of images (when processed serially), or else I dramatically misunderstood something.
* This might be no an issue with this paper per se, but more general- is AP the right metric for this? In my opinion, it isn’t because object discovery is not well-defined (part vs. whole object ambiguity), and therefore, every (possibly relevant) discovered region that is not labeled would be considered as FP and penalize the performance. Using incorrect metrics may hinder progress in this field. Why not evaluating (average) recall wrt. pre-fixed proposal thresholds instead?
* This is also a more general comment: object discovery formulation, as tackled in this paper (and formalized by Vo et al., CVPR’19), is inherently about finding commonly occurring patterns in the image collection. Wrt. the long-tailed distribution of object classes, this will only be applicable to the “head” classes and will dismiss the tail. I am not saying that methods tackling the discovery of frequently occurring objects/patterns are not valuable, but I believe that this venue explicitly encourages discussion of limitations, and this is certainly one of them!



Justification

———————

I think this paper presents a nice contribution for scalable optimization-based object discovery. Besides having solid theoretical ground, I find this paper well explained and experimentally validated and overall find that it should be a nice contribution to this venue.

Post-rebuttal:


———————

This is my summary of my thoughts on the other reviews and rebuttals:

**3zeH, N6wG express doubts on object discovery using a graph-based formulation that finds common patterns based on connectedness analysis:**

I find that this approach to object discovery has already been well-motivated and established in the community (Vo et al., CVPR’19, and ECCV’20). I agree that this is a bit exotic research direction, but I still think it is one worth pursuing, especially given that nowadays, we are collecting large amounts of unlabeled data that “we don’t know what to do with.” Moreover, the novelty of this paper is not in proposing to pose discovery as a combinatorial optimization problem, but rather seeing parallels between methods by Vo et al., and link analysis that admits highly parallelizable algorithms.

**Use of ‘supervised features’ and on the dependability on object proposals (3zeH):**

I believe this work follows the experimental protocol proposed in papers by Vo et al., plus analyses performance using both hand-crafted and deep features.
I think pretty much all discovery methods depend on object proposals (as do several object detectors). I see no issue here.

**Finding commonly-occurring patterns will inherently ignore the “long-tail”** — I’ve had a similar concern as N6wG on this; however, the rebuttal reports a weak correlation with object occurrence frequency. I would like to see a more detailed discussion on this, but either way, I also do believe that the discovery of frequently occurring objects is an open and very challenging problem worthy of new contributions.

**Re-formulation is incremental:** I respectfully disagree with FJwA that establishing this link is of incremental nature; it certainly is obvious in hindsight, but noting the link, re-formulating the objective proposed in Vo et al., and demonstrating experimentally that this yields a highly scalable and parallelizable solution is a solid contribution in my view.

Overall I find this paper relevant and valuable, and I think that the paper should be accepted.

**Time Spent Reviewing:**

4 hours

---

> ### Author Response · Authors · 2021-08-10
> **We thank the reviewer for the helpful comments and suggestions and address the reviewer's concerns below.**
>
> **Q: Besides parallelization, it is also very nice that this formulation is scalable enough that it doesn’t need to limit the number of object proposals per image, which I assume is the main reason behind slightly better performance compared to [Vo et al., ECCV’20]. At the same time, I would also like to note that it would be nice to discuss this experimentally.**
>
> Thank you for pointing this out. It is true that, unlike [67], we are able to use almost all the regions produced by the proposal algorithm (2000 regions per image at most) thanks to the good scalability of our formulation. On average, we have 814 and 850 regions per image on C20K and Op50K, respectively.  We have now evaluated LOD on C20K and Op50K using different numbers of proposals and observed that its performance improves with additional region proposals, notably in the multi-object setting. When a subset of only 100 regions is used, the average precision (AP50) of LOD matches the top performance of [67], and increases with more regions (the full results are shown in the table below). This observation partly explains our better performance compared to [67] (which places a limit on the number of regions for computational reasons) and the benefit of using all region proposals.
>
> | # regions || C20K ||| Op50K ||
> |:--------:|:------:|:----:|:----------:|:------:|:----:|:----------:|
> |           | CorLoc | AP50 | AP@[50:95] | CorLoc | AP50 | AP@[50:95] |
> |     50    |  40.9  |  4.5 |    1.22    |  42.0  | 4.55 |    1.31    |
> |    100    |  44.0  | 5.38 |    1.47    |  43.4  |  5.1 |     1.4    |
> |    200    |  46.5  | 6.13 |    1.71    |  45.6  | 5.83 |    1.61    |
> |    400    |  48.0  |  6.6 |    1.91    |  47.1  | 6.32 |    1.77    |
> |    All    |  48.5  | 6.63 |    1.98    |  48.1  | 6.46 |    1.88    |
>
> **Q: The most problematic aspect of this paper is the very vague definition of what “other top-performing methods cannot handle” (Tab 1.). This is further related to Sec. 4.1, run time discussion, which states that “[32] and [67] did not finish in reasonable time”. What is considered to be a reasonable time? One week (par. “Quantitative evaluation” mentions that)? For example, training neural networks can easily take one week or more, and if a method needs more than a week (2 weeks?) to finish, I would not say that it is not applicable to such a large dataset. I understand that one cannot wait for the optimization to converge for months, but I am really in the dark here what kind of time spans we are talking about, and I would like to see a discussion on this (also the final version of the paper must be more precise here). Also, let me remark here that “reasonable time” will be different in the future with better compute (although it is true that datasets will be growing as well).**
>
> In our experiments, we ran all methods for up to one week (the time limit for a sequential job on our server). After a week, we observed that [67] finished about 18\% of its computation. We agree that it would be important to report more precise times, and will add full timings and results to the final version of the paper if it is accepted.
>
> **Q: It can be seen that optimization-based methods scale linearly (i.e., scale very well). Therefore we can also expect that these methods should be very applicable in the future. I definitely see that the major advantage of the proposed method due to the fact that it is nicely parallelizable, but this aspect is related to the core contribution of this paper. Finally, I do not understand why EB runtime is constant? It should be linear wrt. the number of images (when processed serially), or else I dramatically misunderstood something.**
>
>
> We agree that the optimization-based methods will continue to be applicable and we hope that our exploration into their performance will spur development of other efficient solutions.
>
> It is true that the run time of EB is linear in the number of images, but since processing a single image takes only 1.3s when processed sequentially, its run time is very small compared to the other methods. Therefore, the graph in Figure 2 appears flat for EB. We will clarify this.
>
> **Q: This might be no an issue with this paper per se, but more general- is AP the right metric for this? In my opinion, it isn’t because object discovery is not well-defined (part vs. whole object ambiguity), and therefore, every (possibly relevant) discovered region that is not labeled would be considered as FP and penalize the performance. Using incorrect metrics may hinder progress in this field. Why not evaluating (average) recall wrt. pre-fixed proposal thresholds instead?**
>
>
> Finding the right metric to assess multi-object discovery is a complex issue. The CorLoc metric used in single-object discovery, i.e., the precision when only one region is selected per image, is not appropriate in this context. We use Average Precision (AP), a popular metric for object detection, in the multi-object discovery evaluation to avoid the dependence on the number $m$ of selected regions per image, unlike [67] which computes recall at a fixed value of $m$. We acknowledge that the definition of AP given in our submission is vague and may be misleading. To be more precise, the notion of AP used in all our experiments is correctly defined as the area under the precision-recall curve where precision and recall are computed for each value of $m$ from 1 to $M$ (the maximum number of ground-truth objects per image). As such, this metric accounts for both precision and recall.
>
> Due to the lack of dedicated datasets for object discovery, we evaluate our approach on popular benchmarks for object detection. As pointed out by the reviewer, this protocol is not ideal as there may exist many unlabelled objects that are returned but considered false positives (see Figure 3 for sample examples). Object discovery remains a challenging problem and we hope that our work encourages further research, including new dedicated datasets.
>
>
> **Q: This is also a more general comment: object discovery formulation, as tackled in this paper (and formalized by Vo et al., CVPR’19), is inherently about finding commonly occurring patterns in the image collection. Wrt. the long-tailed distribution of object classes, this will only be applicable to the “head” classes and will dismiss the tail. I am not saying that methods tackling the discovery of frequently occurring objects/patterns are not valuable, but I believe that this venue explicitly encourages discussion of limitations, and this is certainly one of them!**
>
> It is true that, by definition, both the optimization-based existing object discovery methods and the proposed technique are biased toward the frequently occurring patterns and objects. However, we have observed little correlation between LOD’s performance on an object class and the class appearance frequency (corr = -0.09, see figure at https://drive.google.com/file/d/1lAPv4IJa-CUJqF2fMaOCpAVM8M_onsUJ/view?usp=sharing). It could be due to the fact that even though we rank all regions in the image collection at once, we choose objects (based on the ranking) from each image separately. Thus, objects are returned if they stand out and are better connected than other regions in the same image, even if they are objects of a rare class. On the other hand, we have observed that finding small objects is more challenging than finding bigger ones. LOD’s performance on a class correlates well (corr = 0.76) with the average object size (relative to the image size), and the same phenomenon has been observed with supervised object detectors.  We will add a more thorough discussion of limitations to the paper.

---

> > ### Comment · Reviewer_1AuG · 2021-08-31
> > **Thanks for the feedback**
> >
> > I would like to thank the authors for addressing my concerns. I also carefully read your thoughts on this paper and responses, and here is my quick summary:
> >
> > **3zeH, N6wG express doubts on object discovery using a graph-based formulation that finds common patterns based on connectedness analysis:**
> >
> > I find that this approach to object discovery has already been well-motivated and established in the community (Vo et al., CVPR’19, and ECCV’20). I agree that this is a bit exotic research direction, but I still think it is one worth pursuing, especially given that nowadays, we are collecting large amounts of unlabeled data that “we don’t know what to do with.” Moreover, the novelty of this paper is not in proposing to pose discovery as a combinatorial optimization problem, but rather seeing parallels between methods by Vo et al., and link analysis that admits highly parallelizable algorithms.
> >
> > **Use of ‘supervised features’ and on the dependability on object proposals (3zeH):**
> >
> > I believe this work follows the experimental protocol proposed in papers by Vo et al., plus analyses performance using both hand-crafted and deep features.
> > I think pretty much all discovery methods depend on object proposals (as do several object detectors). I see no issue here.
> >
> > **Finding commonly-occurring patterns will inherently ignore the “long-tail”** — I’ve had a similar concern as N6wG on this; however, the rebuttal reports a weak correlation with object occurrence frequency. I would like to see a more detailed discussion on this, but either way, I also do believe that the discovery of frequently occurring objects is an open and very challenging problem worthy of new contributions.
> >
> > **Re-formulation is incremental:** I respectfully disagree with FJwA that establishing this link is of incremental nature; it certainly is obvious in hindsight, but noting the link, re-formulating the objective proposed in Vo et al., and demonstrating experimentally that this yields a highly scalable and parallelizable solution is a solid contribution in my view.
> >
> > Overall I find this paper relevant and valuable, and I think that the paper should be accepted.

---

### Official Review · Reviewer_N6wG · 2021-07-16

**Rating:** 5
**Confidence:** 3

**Summary:**

This paper deals with object discovery problems (single object discover as well as multiple object discover). It proposed to solve the challenge as a ranking problem. It first generates object proposals on all images in the dataset. Then object proposals are ranked based on their similarity to regions in other images. The ranking task in the paper is performed with PageRank. The experiment shows it achieves promising results on the COCO and OpenImage dataset.

**Limitations And Societal Impact:**

I wonder whether performing ranking based on object proposal similarity with other regions in the dataset leads to an ill configuration of the unsupervised object detection problem. It encourages discovery of frequently appearing objects, when suppressing objects rarely appear.  I would be great if the paper could provide the performance over object categories with different   instance numbers, respectively.

**Main Review:**

Originality: Utilizing ranking for object discovery seems original to me.
Clarity: The paper lacks motivations/discussions of the proposed technology, making it hard to follow why a mathematical method is employed in the framework. Adding intuitive discussions about why ranking is a good technical solution for object discovery would improve the quality of the paper.
Quality: The paper is technically correct to me. Graph connection density is definitely an implication about the frequently appearing objects. However, it corresponds to discovering the frequent objects. The paper could add more discussion about whether the technology is a good fit for the task of object discovery which looks to discover single/all foreground objects.
Significance: The paper is of interest of the community.

**Time Spent Reviewing:**

5

---

> ### Author Response · Authors · 2021-08-10
> **We thank the reviewer for the helpful comments and suggestions and address the reviewer's concerns below.**
>
> **Q: The paper lacks motivations/discussions of the proposed technology, making it hard to follow why a mathematical method is employed in the framework. Adding intuitive discussions about why ranking is a good technical solution for object discovery would improve the quality of the paper.**
>
> As discussed in Section 2.1, it is natural to cast unsupervised object discovery (UOD) as the task of finding repetitive visual patterns in image collections (what other notion of “object” would be available without any supervision?). Recent approaches (e.g., [66,67]) to UOD formulate it as a combinatorial optimization problem, with the corresponding computational limitations. The motivation behind our work is to formulate UOD as a simpler graph-theoretical problem with a more efficient solution, where objects correspond to well-connected nodes in a graph whose nodes are region proposals (instead of images in [66,67]), and edges are weighted by region similarity and objectness. In this scenario, finding object-proposal nodes is now a ranking problem where the goal is to rank the nodes based on whether they represent objects. Another work with a similar motivation that uses ranking (of images) is VisualRank [27], where an image search problem is solved by applying PageRank to the pairwise similarity matrix in an image collection.
>
> From another perspective, given a set of region proposals an image, discovering objects means finding the most object-like regions. This naturally amounts to ranking them according to their ``objectness’’. In fact, [66,67] implicitly rank regions in each image by assigning them a binary score ($x_{ij}$). Different from them, we rank all regions in the image collection by assigning them positive real scores. We show that this relaxation allows more effective and scalable solutions to UOD. We will clarify these points in the paper if it is accepted.
>
> **Q: Graph connection density is definitely an implication about the frequently appearing objects. However, it corresponds to discovering the frequent objects. The paper could add more discussion about whether the technology is a good fit for the task of object discovery which looks to discover single/all foreground objects.**
>
> Similar to VisualRank [27], we assume that a high similarity score between a pair of proposals is an indicator of whether the corresponding two proposals may correspond to a common foreground object. We use the PHM algorithm to compute the similarity score in our work. PHM is a probabilistic algorithm that uses a voting procedure among the pairs of regions, and given that regions are concentrated in foreground areas, the resulting similarity matrix favors discovering foreground objects. This enables our ranking formulations to find visual patterns that are both repetitive and object-like. Further discussion of PHM can be found in [8].
>
>
> **Q: I wonder whether performing ranking based on object proposal similarity with other regions in the dataset leads to an ill configuration of the unsupervised object detection problem. It encourages discovery of frequently appearing objects, when suppressing objects rarely appear. It would be great if the paper could provide the performance over object categories with different instance numbers, respectively.**
>
> It is true that, by definition, both the existing optimization-based object discovery methods and the proposed technique are biased toward frequently occurring patterns and objects. However, we have computed LOD’s performance by object category on C20K dataset and observed little correlation between the performance on an object class and its appearance frequency (the corresponding correlation is only -0.09, see figure at https://drive.google.com/file/d/1lAPv4IJa-CUJqF2fMaOCpAVM8M_onsUJ/view?usp=sharing). A possible explanation is that even though we rank all regions in the image collection at once, we choose objects (based on the ranking) from each image separately. Therefore, regions can be selected as objects if they stand out more from the background and are better connected in the graph than other regions in the same image, even if they represent objects of a rare class. On the other hand, we have observed that finding small objects is more challenging than finding bigger ones and the performance of LOD on a class correlates well (correlation score of  0.76) with average object size (relative to the image size). This is not surprising as the same phenomenon has been observed with supervised object detectors. As suggested by the reviewer, we will add this discussion to the paper.

---

### Official Review · Reviewer_3zeH · 2021-07-17

**Rating:** 6
**Confidence:** 2

**Summary:**

The authors propose a scalable approach (compared to existing ones)
for the 'unsupervised object discovery' task, based on graph and link
analysis methods. They show on several datasets and on the two tasks
of single and multiple object discovery, that their method is either
substantially more scalable and/or has better accuracy.

**Limitations And Societal Impact:**

Yes.

**Main Review:**


Summary of review:

-- Interesting approach and problem, but I couldn't convince myself that the idea of using a graph could
improve results so much:
Why should the overall idea of link-analysis/ranking by 'authority/hubs' idea, behind the particular objective and their particular  algorithms,
substantially help for this problem? it would be useful  to give more
 intuition (the authors do try to explain/motivate in a few cases)... (see below for more details on this ..)

-- use of 'supervised features' defeats the purpose/motivation  of the 'unsupervised'  task. The
 authors have a nice section on using recent work that doesn't use
 supervised features ... perhaps  they should start with that and make that more central
 (because it appears that they are testing on datasets where labels
 overlap significantly between where the features were trained on (eg
 imagenet) and those datasets ) ..

-- sensitivity analysis to various params (eg max number of neighbors or
 random perturbations?): do they do this in the paper?

-- a number  of clarity (presentation) issues. please see below.

---------------
More details (the numbers refer to line numbers):

why should this work? it would be useful to give more intuition..

in particular: objectness based on votes of similarity of neighbors.. it seems that
if two proposed regions are very similar we may get high objectness
for both regions.. and/or (more generally) if there is a large
(semi)clique of very similar regions in different images, we could get
high (in paper, it's called y) scores..  So what if we replicated each
image k times in the corpus.. (some sufficiently large replication factor above max
number of neighbors in the optimization) would that that foil this
approach?


61: do you mean link 'regions' vs 'images' in 'and link images that
contain similar objects..' (since an image can contain multiple
regions/objects?)

62: does every image have exactly r proposals?  It's not clear whether
this is variable or fixed.  It would be good to give clear
(mathematical) definitions too (instead of solely relying on English).
If r is fixed, what happens to images that should get no or only one or
 fewer than r proposals (do we create dummy proposals for such?). Good to make
this clear.
...


66: not sufficiently clear.. what does 'correspond to visual content
shared' mean?  what does 'share' mean? This definitely  needs to be clarified,
since this is problem definition (see below too).

One clarification is wrt to number, assuming 'sharing similar visual
content' means same objects: do you mean, some of the neighbors have
the same object, or *all* the neighbors, or at one least one, etc:
"Similarly, let x_kp ∈ {0,1} for p = 1,2,...,n and k = 1,2,...,r be
indicator variables such that x_kp = 1 when region proposal k of image
p corresponds to visual content shared with its neighbors and let xp =
(x1p , . . . , xrp )T ."

- this approach depends very largely on accuracy of region
  proposals.. (very much depends on..)

- minor: as an aside: why tao and rho are constants.  It seems making the
  number of neighbors image and region dependent, perhaps based on a
  similarity threshold, would be better... The nature of the two
  parameters seem different: one (max number of regions per image) is
  property of the data and region detection algorithms used, while the
  other (max number of neighbors allowed) seems to be a meta
  parameter, provided by the user.


- minor: again, the English description of tao is not very clear.  It
  likely means over all the images, take the maximum.  if written in
  mathematics, in addition or instead of, just saying it in English,
  it would be clearer.


- 243: (in multiple object regime) Does the user specify M or
 otherwise, in an unsupervised problem, what can M be?? (I am thinking
 in their experiments, they know what M is from the datasets..)  in
 "we return up to M regions per image, where M is the maximum number
 of objects in any image in the dataset. "


- 311: doesn't defeat the purpose of 'unsupervised' object discovery to
use image-net trained models (even for features??).. in
"Self-supervised features vs. supervised features. LOD and all of the
optimization-based base-lines [32, 67, 71] rely on a VGG [60]-based
classifier trained on ImageNet [11]."

- 400: '87x'   in   "..allows us to scale up UOD to the OpenImages [35] dataset
(Op1.7M) with 1.7M images, 87 larger
than datasets considered in the previous state-of-the-art technique [67], and outperforms

- multi objective performances seem very low .. perhaps some
  discussion of that?

=============

After seeing author replies and other reviewers.

I have read the reviews and authors' responses.   Thank you all. Given the authors' responses, I have increased
my rating to leaning to accept, as I trust the authors will address a main issue, use of supervised vs unsupervised features, well, and considering the other author responses to various issues I raised, and the other reviewers' feedback (and the presentation issues can be mostly addressed). Thank you.



**Time Spent Reviewing:**

4.5

---

> ### Author Response · Authors · 2021-08-10
> **We thank the reviewer for the helpful comments and suggestions and address the reviewer's concerns below.**
>
> **Q: Why should the overall idea of link-analysis/ranking by 'authority/hubs' idea, behind the particular objective and their particular algorithms, substantially help for this problem?**
>
> As discussed in Section 2.1, it is natural to cast unsupervised object discovery (UOD) as the task of finding repetitive visual patterns in image collections (what other notion of “object” would be available without any supervision?). Recent approaches (e.g., [66,67]) to UOD formulate it as a combinatorial optimization problem, with the corresponding computational limitations. The motivation behind our work is to formulate UOD as a simpler graph-theoretical problem with a more efficient solution, where objects correspond to well-connected nodes in a graph whose nodes are region proposals (instead of images in [66,67]), and edges are weighted by region similarity and objectness. In this scenario, finding object-proposal nodes is now a ranking problem where the goal is to rank the nodes based on whether they represent objects. Another work with a similar motivation that uses ranking (of images) is VisualRank [27], where an image search problem is solved by applying PageRank to the pairwise similarity matrix in an image collection.
>
> The ranking formulation leads to very efficient solutions from link analysis, including the quadratic formulation (Q) and Pagerank. This allows us to use all the proposals, instead of a subset as in [67], which explains in part LOD’s superior performance at discovering multiple objects. Indeed, in an experiment suggested by reviewer 1AuG, we have observed that LOD’s multi-object discovery performance increases as more regions are considered. On C20K and Op50K datasets, it obtains an AP50 (5.38 and 5.1 resp.) comparable to [67]’s (5.18 and 4.98 resp.) when using only 100 regions per image but reaches significantly better performance when using 200 regions (6.13 and 5.83 resp.) or all regions (6.63 and 6.46 resp.).
>
> There is a price to pay of course for focusing on the proposal graph instead of the image graph: Our approach foregoes the explicit recovery of the global image collection structure (which images contain the same objects?). However, we demonstrate in the paper that this structure can be recovered later as a post-processing step. We will clarify these points in the paper if it is accepted.
>
> **Q: use of 'supervised features' defeats the purpose/motivation of the 'unsupervised' task. The authors have a nice section on using recent work that doesn't use supervised features, perhaps they should start with that and make that more central**
>
> We thank the reviewer for pointing this out. We are, as far as we know, indeed the first to use self-supervised features to create a fully unsupervised pipeline for UOD. However, since state-of-the-art approaches to UOD report results using supervised features, we have used these features as well in our comparisons. After the submission deadline, we have conducted experiments with self-supervised features on the large Op1.7M dataset and again obtained good performance (CorLoc:49.4, AP50:6.28, AP@[50:95]:1.86). On this dataset, self-supervised features actually fare better than supervised ones (compare to Table 1 in the submission) in single-object discovery and establish a new state-of-the-art performance. We will add these results to Table 1, along with additional discussion on the use of self-supervised features, in order to highlight these points.
>
> **Q: sensitivity analysis to various parameters?**
>
> We have presented our analysis on the influence of the damping factor $\beta$ in PageRank and the scalar $\alpha$ used to select reliable object candidates in LOD in the supplemental material. Following the reviewer’s suggestion, we assess below on the C20K and Op50K datasets the sensitivity of LOD to $N$, the number of initial image neighbors, and to $\gamma$, the parameter controlling the strength of the small perturbation added to the score matrix $W$. For $N$, we have tried different values from 100 to 500 and found that the performance improves only slightly when more neighbors are considered. However, the computational cost increases linearly with $N$ and we find that using 100 neighbors is a good compromise for our datasets. As discussed later, the number of neighbors might need to be changed in case of (undetected) near duplicates or in the case of videos (with successive, highly similar frames). For $\gamma$, we have varied its value in {$10^{-6}, 10^{-5}, 10^{-4}, 10^{-3}, 10^{-2}$} and found that the performance does not vary much ($ < 0.5$\%) when $\gamma \leq 10^{-4}$ and slightly degrades when $\gamma \geq 10^{-3}$. This shows that LOD is insensitive to $\gamma$ as long as it is small enough.
>
> **Q: objectness based on votes of similarity of neighbors, it seems that if two regions are very similar we may get high objectness for both and/or (more generally) if there is a large (semi)clique of very similar regions in different images, we could get high scores. So what if we replicated each image k times in the corpus, would that foil this approach?**
>
> Indeed a large number of replicates, in the limit larger than the number of neighbors used in the optimization, would degrade performance considerably. Note, however, that (1) there are no replicates in the standard benchmarks we use, and (2) in the perhaps more realistic scenario where images would be collected from the web with some, perhaps many exact or close image replicates, we could use an effective near-duplicate image detection method, e.g., (Dong et al., 2012), as a pre-processing step. A similar idea could be used for video datasets. In the video case, random, or better, representative keyframes can be extracted and used as input to LOD.
>
> Reference:
> Dong, Wei, et al. "High-confidence near-duplicate image detection." ACM ICMR. 2012.
>
> **Q: this approach depends very largely on accuracy of region proposals**
>
> As is the case for many methods for object detection-related tasks, region proposals are an important component in our pipeline and influence the performance. We have shown in the paper that LOD obtains good results with the proposal algorithm from [67]. However, any unsupervised proposal algorithm can be used in LOD and its performance can be further improved with better proposals.
>
> **Q: 61: do you mean link 'regions' vs 'images' in 'and link images that contain similar objects.' (since an image can contain multiple regions/objects?)**
>
> In line 61, we describe the approach taken by [67]. It attempts to link images that contain similar objects by optimizing binary variables $e_{pq}$ which represent whether images $p$ and $q$ contain at least one common object.
>
> **Q: 62: does every image have exactly r proposals? It's not clear whether this is variable or fixed. It would be good to give clear (mathematical) definitions too. If r is fixed, what happens to images that should get no or only one or fewer than r proposals?. **
>
> We assume that each image has exactly $r$ proposals for the sake of simplicity. In practice, we choose up to 2000 proposals per image so that we take essentially all the proposals returned by the proposal algorithm (it is not required that every image has the same number of proposals). On the other hand, we always add the entire image rectangle as a proposal to guarantee that each image has at least one proposal. We will clarify these points in the final version.
>
> **Q: 66: not sufficiently clear, what does 'correspond to visual content shared' mean? what does 'share' mean? This definitely needs to be clarified, since this is problem definition.**
>
> By “region proposal $k$ of image $p$ corresponds to visual content shared with its neighbors” we meant “region proposal $k$ of image $p$ is an object-like region in image p that is similar to an object-like region in one of the neighbors of image p”. This will be clarified in the paper.
>
> **Q: minor: as an aside:why tao and rho are constants.**
>
> We do not use $\tau$ and $\nu$ in our approach (this section describes the previously proposed formulation (C) described in [66,67]). There, $\tau$ and $\nu$ serve as constraints that force the model to find the best neighbors and regions.
>
> **Q: 243: Does the user specify M or otherwise, in an unsupervised problem, what can M be in "we return up to M regions per image, where M is the maximum number of objects in any image in the dataset."?**
>
> We follow another work on object discovery on geometric shapes [43] and assume in our submission that $M$ is known during evaluation. It is 77 in C20K, 90 in C120K, 543 in Op50K and 745 in Op1.7M. In a real application, one would of course use a rough "budget estimate" of the upper bound on how many objects per image one may try to detect.
>
> **Q: multi objective performances seem very low, perhaps some discussion of that?**
>
> Finding the right metric to assess multi-object discovery is a complex issue. The CorLoc metric used in single-object discovery, i.e., the precision when only one region is selected per image, is not appropriate in this context. We use Average Precision (AP), a popular metric for object detection, in the multi-object discovery evaluation to avoid the dependence on the number $m$ of selected regions per image, unlike [67] which computes recall at a fixed value of $m$. We acknowledge that the definition of AP given in our submission is vague and may be misleading. To be more precise, the notion of AP used in all our experiments is correctly defined as the area under the precision-recall curve where precision and recall are computed for each value of $m$ from 1 to $M$ (the maximum number of ground-truth objects per image). Since the precision decreases significantly with increasing $m$, the performance numbers appear much smaller than in the single-object case. As noted in the paper, UOD is still a very challenging task due to the lack of supervision and the multi-object discovery performance is relatively low for all methods.

---

### Decision · Program_Chairs · 2021-09-27

**Decision:**

Accept (Poster)

**Comment:**

The formulation of unsupervised object detection as a PageRank problem is neat, and allows to scale up to significantly larger datasets.
The paper is accepted, but authors are encouraged to better motivate the use of supervised features (and/or highlight the experiments with self-supervised features more).